# Genetic Control of Avian Migration: Insights from Studies in Latitudinal Passerine Migrants

**DOI:** 10.3390/genes14061191

**Published:** 2023-05-29

**Authors:** Aakansha Sharma, Sayantan Sur, Vatsala Tripathi, Vinod Kumar

**Affiliations:** 1IndoUS Center in Chronobiology, Department of Zoology, University of Lucknow, Lucknow 226007, India; 2Department of Zoology, Dyal Singh College, University of Delhi, Delhi 110003, India; 3IndoUS Center in Chronobiology, Department of Zoology, University of Delhi, Delhi 110007, India

**Keywords:** bird, brain, gene expression, heritability, migration, seasonal

## Abstract

Twice-a-year, large-scale movement of billions of birds across latitudinal gradients is one of the most fascinating behavioral phenomena seen among animals. These seasonal voyages in autumn southwards and in spring northwards occur within a discrete time window and, as part of an overall annual itinerary, involve close interaction of the endogenous rhythm at several levels with prevailing photoperiod and temperature. The overall success of seasonal migrations thus depends on their close coupling with the other annual sub-cycles, namely those of the breeding, post-breeding recovery, molt and non-migratory periods. There are striking alterations in the daily behavior and physiology with the onset and end of the migratory period, as shown by the phase inversions in behavioral (a diurnal passerine bird becomes nocturnal and flies at night) and neural activities. Interestingly, there are also differences in the behavior, physiology and regulatory strategies between autumn and spring (vernal) migrations. Concurrent molecular changes occur in regulatory (brain) and metabolic (liver, flight muscle) tissues, as shown in the expression of genes particularly associated with 24 h timekeeping, fat accumulation and the overall metabolism. Here, we present insights into the genetic basis of migratory behavior based on studies using both candidate and global gene expression approaches in passerine migrants, with special reference to Palearctic-Indian migratory blackheaded and redheaded buntings.

## 1. Introduction

Avian migration is a regular, seasonal, large-scale movement of a population between fixed breeding and wintering geographical locations [1]. Billions of birds undertake twice-a-year, migration journeys between breeding and wintering regions that, for many species, lie across the latitudinal gradient. These birds breeding in the northern hemisphere fly southwards in autumn to winter, and they begin their return flight northwards in spring to timely reach their homes (i.e., breeding ground). There are, however, species differences in the migratory distance and, perhaps because of the intraspecific competition, in the direction and location of winter grounds [2,3,4].

Three experimental approaches have been generally used to study the genetic control of migratory behavior in birds [5]. These approaches aim to examine if a migratory trait is genetically or environmentally determined, to predict the adaptability and evolution rate of a migratory trait using a quantitative genetic approach, and to identify specific genes or gene sets involved in controlling the expression of the migratory behavior. The first approach centers mainly around identifying the components of a migratory behavior having a genetic basis, and involves translocation, cross-fostering or cross-breeding experiments (for details, please see [6,7]). The second approach involves a number of quantitative genetic methods, e.g., the parent–offspring regression and full sibling correlations aimed at deciphering the adaptability and evolution of the migratory behavior trait [5]. The third and more recent approach focuses on the idea of the “migratory gene package” that controls migration-linked changes in the morphology, behavior and physiology [8].

## 2. Migration: A Heritable Seasonal Behavior

A successful migratory journey requires the occurrence of sequential changes at several levels with great precision. It is therefore important that a migratory trait has been genetically shaped and adapted to the external environment. Over the years, this proposition has been experimentally tested, as described briefly below.

### 2.1. An Innate Migratory Template

Early in the 20th century, the experimental evidence came for an innate nature and inheritance of avian migratory behavior [9,10,11]. A more actual genetic basis of migration was published in the 1990s from a series of cross-breeding experiments of a non-migratory population (of Cape Verde Islands) with a migratory population (from southern Germany) of blackcaps (*Sylvia atricapilla*); ~40% of F1 hybrids were migratory [7,12]. Most interestingly, the migratory urge seemed genetically transmissible, and the urge could be further manipulated in magnitude by selective breeding experiments involving a partially migratory population [12]. All hybrids were migratory, suggesting that the migratory urge was determined by a multi-locus system having a reaction threshold in the face of environmental changes [7,12]. Further, common garden experiments on blackcaps have shown the genetic basis of both the distance [13] and direction of seasonal migrations [14].

There is also good evidence to show that a migrant can complete its migratory travel without guidance from experienced conspecifics, as concluded from a study on common cuckoos (*Cuculus canorus*). Satellite tracking of migrating cuckoos revealed that juveniles had initiated migratory travel later than the adults, and they traveled via a straighter and faster route to reach their African wintering grounds, perhaps as independently guided by their innate migration programs [15].

### 2.2. Heritable Migratory Activity Pattern

Both spatiotemporal migration program and migratory activity are heritable [7,12,16]. A study on parent–offspring regression and full sibling correlation as calculated from migratory activity of 280 southern German blackcaps from 69 families revealed that variability in the migratory behavior had come majorly from the genetic difference and with little, if any, contribution from the environment [17]. The pattern of migratory activity also in migratory garden warblers (*Sylvia borin*) was fairly insensitive to food availability or weather conditions, although a very long photoperiod could modify the amount of night activity [18,19,20]. Likewise, the migratory direction follows a dominant inheritance pattern, as shown by a genome-wide study on nearly identical two willow warbler (*Phylloscopus trochilus*) subspecies that have drastically different migratory routes. The subspecies *P. t. trochilus* migrates southwest from western European breeding to western African breeding grounds, while *P. t. acredula* migrates from southeast from northern and eastern Europe to eastern and southern Africa [21,22]. The single nucleotide polymorphisms (SNPs) of juveniles revealed that the hybrids of these two subspecies followed an intermediate route during the autumn migration [23]. Very recently, Sokolovskis et al. [24] discovered that the dominance to southwestern migratory direction in *P. t. trochilus* was associated with the *trochilus* allele at inversion polymorphisms on chromosomes 1 (InvP-Ch1), and to southeastern migratory direction in *P. t. acredula* was associated with migration associated repeat block (MARB-a); MARB-a seemed to have an epistatic suppressive effect on the InvP-Ch1 [24].

### 2.3. Heritable Arrival Dates

An important aspect of migration is the timing of arrival at the breeding site in order to have prior access to the prime breeding habitat and a better mate choice [25]. A decline in breeding success among late arrivals at breeding grounds has been documented for several migrant species [26,27,28,29,30,31]. Interestingly, this trait could also be heritable. Tarka et al. [32] studied the quantitative genetics of arrival dates at the breeding ground using data sets collected over 20 years (multigenerational pedigree) for both sexes of great reed warblers (*Acrocephalus arundinaceus*). There was 16.4% heritability for the arrival dates as well as directional selection for early arrivals in both sexes acting through the reproductive success [32].

Studies over the last three decades also suggest a correlation of rising mean spring temperature with the arrival date and hence the early start of reproduction [33]. Given the high heritability of the arrival dates [32], a strong response to climate change-induced selection (a micro-evolutionary change rather than the phenotypic response) can be expected [34]. Indeed, the results from Swedish data on spring arrival dates of migratory birds from the last 140 years suggest that an increased temperature possibly led to advanced spring arrival dates, and that the spring migration-linked phenologies have shown bigger changes in short-distance than those in the long-distance migrants [35].

## 3. Unraveling the Genetic Control of Migratory Behavior

### 3.1. Gene Polymorphism

A comparative study of gene polymorphism in an experimentally simulated (e.g., a reference experiment) or in naturally occurring (e.g., resident vs. migratory population) functional states can reveal the genetic basis of a trait, such as seasonal migration.

Using gene polymorphism as a molecular approach, few studies have documented the association of a single gene polymorphism with the migratory phenotype, while few others have failed to show such an association. For example, there was a significant correlation of *Adcyap1* (adenylate cyclase activating polypeptide 1) gene polymorphism with the migratory restlessness in blackcaps; birds with longer *Adcyap1* allele exhibited higher *Zugunruhe* [36]. A similar *CLOCK* (circadian locomotor output kaput) gene polymorphism was found associated with enroute stay (stopover) times (a longer *CLOCK* allele suggests delayed departure from Mediterranean island stopover sites) in trans-Saharan migratory common nightingales (*Luscinia megarhynchos*), European pied flycatchers (*Ficedula hypoleuca*), tree pipits (*Anthus trivialis*) and whinchats (*Saxicola rubetra*) [37,38]. At the same time, Peterson et al. [39] found no consistent association of *Adcyap1* or *CLOCK* gene allele lengths with the migratory status in a study of 15 different populations across two subspecies of *Junco* ranging from sedentary to long-distance migrants. Notably, however, the long-distance migratory Juncos had longer *CLOCK* alleles on average and, similar to captive blackcaps, the *Adcyap1* allele length showed a positive correlation with the migratory restlessness among individuals of one of the two *Junco* populations studied, indicating the association of *Adcyap1* gene with migratory propensity within or between certain populations only [38]. The *CLOCK* gene polymorphism was also not associated with the migration phenotype in bar-tailed godwit (*Limosa lapponica baueri*) [40]. 

Similarly, a single nucleotide polymorphism of *Vps13a* (vacuolar protein sorting 13 homolog A) gene was found to be associated with migration directionality in two very closely genetically related *Vermivora* warbler species with similar breeding sites in North America and vastly different wintering sites in different geographical directions—golden-winged warbler (*Vermivora chrysoptera*) winters in Central America and blue-winged warbler (*V. cyanoptera*) winters in South America [41,42]. There was a reduced sequence variation in the *Vps13a* gene region among South America wintering warblers, indicating the likelihood of natural selection on this gene locus [43]. Likewise, a genome sequencing study of four different populations of peregrine falcons (*Falco peregrinus*) indicated the association of *Adcy8 (adenylate cyclase 8)* gene with population level differences in the migratory distance [44]. 

Notably, blackcaps have emerged as an experimental model system to study the genetics and epigenetics of migration [45]. To trace the evolutionary history of migration in blackcaps, for example, Delmore et al. [46] using high-throughput sequencing technologies sequenced the genome of 110 blackcaps from populations exhibiting differences in the autumn migration. Along with a revelation that the divergence began about 30,000 years ago, a small set of genes was found to code for differences in their migratory behavior. The genetic variations occurred in the regulatory region, not in gene sequence, suggesting the possibility for the occurrence of rapid changes in the migratory behavior [46]. A study on migratory American kestrels (*Falco sparverius*) further suggests this [47]. In kestrels that were captured during the autumn migration, the genetic variation in gene loci that modulate the internal biological clock (e.g., *Top1*, *Phlpp1*, *Cpne4* and *Peak1* genes) accounted for an intra-population existence of both early and late migratory chronotypes, supporting the argument that the variation in regulatory gene regions was responsible for the variation in migration phenotypes [47].

### 3.2. Gene Transcription

A migratory phenotype is expressed for a specific purpose and for a defined time period. Additionally, the two seasonal migrations (to-and-fro movement between breeding and wintering grounds) differ in several ways. For example, spring and autumn migrations differ in context: the spring travel is for the timely arrival at the breeding grounds (hence there is a stronger reproductive drive), whilst the autumn travel is essentially for escaping harsh winter conditions at breeding grounds and for finding adequate food resources for spending the winter season. This is reflected in the higher speed, longer nocturnal flights and shorter stopovers during spring than during the autumn migration [48,49,50]. Birds are also in different physiological states prior to the onset of the two migrations: they are sensitive to long-day photostimulation in spring and refractory to it in the autumn (Figure 1, [51]). Such differences in the phenotype are possibly the results of differential regulatory molecular strategies that can be deciphered by examining differences in the gene expression pattern between different seasonal states. However, studying the expression pattern using a candidate gene or global gene expression approach is challenging as it can vary significantly across the day and/or the year. It is, therefore, important that the samples for gene expression studies are collected at the same time of the day in different seasonal states. It is also desired to carry out gene expression assays using an appropriate tissue that answers a specific question (e.g., brain tissue for regulation, and liver and flight muscles for the metabolism), and to avoid cross-tissue comparisons.

In recent years, several gene expression studies have been conducted in long distance Palearctic—Indian migratory emberizid finches, the blackheaded bunting, *Emberiza melanocephala* and redheaded bunting, *Emberiza bruniceps*. Some of the results are described in the following headings.

#### 3.2.1. Genes Involved in Photoperiodic Induction Pathway

The annual change in the relative length of daily light and dark periods away from the equator or in the relative intensity of daytime intensity around the equator provides a robust time cue for appropriate timing of behavior and physiology during the year [57,58]. The light information relevant for photoperiod-induced physiology to a large extent is perceived by hypothalamic deep brain photoreceptors (DBPs) in birds [59]. In particular, DBPs containing photopigments OPN5 (opsin 5, also called neuropsin and expressed in CSF-contacting neurons of the paraventricular organ [60]) and VA-opsin (vertebrate ancient opsin, expressed in anterior and medial hypothalamus [61]) have been linked to the neuroendocrine signaling of the external photoperiodic information. Nakane et al. [60] discovered that photic information collected in OPN5 neurons is translated via G-protein-coupled receptors into a biological action in the external zone of the median eminence, juxtaposed to pars tuberalis (PT, the site of thyroid stimulating hormone-β subunit, TSH-β release). TSH-β from PT thyrotrophs binds to its receptors in the tanycytes (ependymal cells) lining the ventrolateral walls of the third ventricle [62], and results in the expression of *Dio2*, a gene that encodes for type 2 deiodinase, DIO2, the thyroid hormone activating enzyme via the Tsh-receptor-cAMP signaling pathway [63]. DIO2 catalyzes the intracellular enzymatic conversion of the prohormone thyroxine (T4), an inactive form, to its biologically active form 3,5,3′-triiodothyronine (T3) which regulates gonadotropin-releasing hormone (GnRH; a hypothalamic peptide) release in long day breeders [63]. 

In a series of experiments, Kumar and his colleagues measured the expression of genes involved in the thyroid hormone-responsive pathway in photoperiodic induction of migratory phenotype in buntings. A study on redheaded buntings reported a significant increase after exposure to a single long day in mRNA levels of *Pax6* (paired box 6) involved in the activation of *Eya3* (Eye absent 3, important for photoperiodic induction), *Tshβ*, *Dio2* and *Rhodopsin* genes [64]. A similar significant increase in the mRNA expression of *Tshβ* and *Dio2* and a decrease in *Dio3* level was found by hour 18 of the first long day in migratory blackheaded bunting (Figure 2, [65]). Interestingly, the peak mRNA expression of *Tshβ* and *Dio2* occurred during hour 14–16 of the day, consistent with the circadian rhythm in photo-inducibility of the photoperiodic response system in buntings [66].

#### 3.2.2. Genes Involved in Temperature Sensitive Pathways

The accompanying feature of photoperiod length is the temperature change. So, temperature can affect the photoperiodic induction of phenologies linked with migration (e.g., body fattening and migratory night-time restlessness, *Zugunruhe*). In an initial experiment on migratory blackheaded bunting in which birds were exposed for four weeks to near-threshold photoperiods at an ambient temperature of 22 °C and 35 °C, there was a reduced body fattening, attenuated activity level and delayed onset of *Zugunruhe* at the higher temperature [69]. Trivedi et al. [70] further tested temperature effects on thyroid hormone-responsive genetic pathways in migratory redheaded buntings subjected to 22 °C and 38 °C temperatures. The *Tshb*-*Dio2*-*Gnrh* gene pathway was upregulated at a high temperature (38 °C), while expression of the *Trpm8* gene (encoding for transient receptor potential cation) was found to be high at 22 °C, compared to 38 °C (TRPM8 is known to be activated in response to ∼8–25 °C temperature; [71]). In a similar experiment also on migratory blackheaded buntings, both cutaneous and hypothalamic expressions of the *Trpv4* gene which senses warm temperatures (∼27–37 °C) were upregulated at 35 °C (Figure 2, [67]).

#### 3.2.3. Genes Involved in Nutrition Sensitive Pathway

Birds accumulate fat (fuel), which is drastic even in small birds and in those that are principally nocturnal migrants [72]. In fact, the subcutaneous fat stores are the best predictor for the duration of the migratory flight and stopovers: birds that have accumulated more fat are more likely to depart sooner than the lean birds that have not accumulated enough fat yet, and they also stay for a shorter duration at the stopover sites [73,74,75].

The fat deposition (i.e., fueling) is largely by hyperphagia, for which both physiological and molecular underpinnings are largely unknown. At this time, the best studied approach with respect to fueling is via the unraveling of mechanisms involved in hunger and feeding, i.e., food intake, which to a large extent is influenced by the energy status and nutrient availability. The hypothalamic arcuate nucleus plays a critical role via homeostatic balance of the activity of orexigenic (neuropeptide Y and agouti-related protein, NPY/AgRP-expressing) and anorexigenic (pro-opiomelanocortin and cocaine- and amphetamine-regulated transcript, POMC/CART-expressing) neurons [76].

In a study related to changes in brain peptides associated with energy homeostasis, Surbhi et al. [77] found seasonal life history dependent variation in the expression level of NPY in DMH, suggesting its role in the food intake and energy balance in migratory redheaded buntings. Likewise, Agarwal et al. [78] found an increased CART but not NPY levels in the ventral tuberal division (inferior hypothalamic nucleus and infundibular nucleus) of redheaded buntings exposed to increasing photoperiods. Further, the 24-h expression pattern of *Npy* in the hypothalamus showed seasonal alterations, with a loss of a significant daily rhythm in the photoperiod-induced migratory state. The high amplitude *Npy* daily rhythm during the winter non-migratory state with peak lying at the end of the dark phase possibly indicated a long-night starvation under winter short days [79].

A closely related aspect of migration is how migrants decide that they are ready to continue their journey, i.e., stays at the stopover sites. In a study, Goymann et al. [74] studied the importance of ghrelin in stopover decisions of wild migratory garden warblers. Ghrelin is a peptide hormone which is secreted from the proventriculus (the glandular part of the stomach) and other organs of the digestive tract and connects the nutritional state to control centers in the brain. The authors found a positive correlation of ghrelin concentrations with fat stores, and warblers administered with ghrelin decreased their food intake and increased their drive to continue migration. 

#### 3.2.4. Genes Involved in Timekeeping

Endogenous clocks enable a long-lived species to identify the time of the year to switch on and switch off its internal mechanisms, in order to synchronize behavior and physiology with the favorable season. In a photoperiodic species, the internal clock is sensitive to daily changes in the photoperiod. The photoperiodic clock shows features that are consistent with a circadian (*circa* = about, *dies* = day) rhythm [80]. Each day, this clock is envisaged as passing through a period of photoinducibility (=the photoinducible phase), which occurs some 12 h after dawn and coincides with an increasing light period in spring. This leads to photoperiodic induction of gene cascade involved in the thyroid hormone responsive pathway [66], and results in the development of the migratory phenotype, namely hyperphagia, fat accumulation and *Zugunruhe* in captive migratory birds [58]. This seems a plausible explanation for the development of the spring migratory phenotype but does not explain the development of the autumn migratory phenotype in response to a decreasing photoperiod. 

An endogenous circadian clock controls the daily alteration between the diurnal hopping and feeding behaviors and the nocturnal appearance of *Zugunruhe*. When *Zugunruhe* is expressed, the free-running period of locomotor activity lengthens [81,82]. The removal of the pineal gland, the dominant avian circadian pacemaker, results in the disruption of daily alterations between *Zugunruhe* and diurnal hopping and feeding behaviors in migratory white-throated sparrows [83]. In migratory redheaded buntings, the absence of pineal gland did not affect the development of long-day induced *Zugunruhe* but decayed the circadian rhythm in *Zugunruhe* as well as in clock genes (*Bmal1, Clock, Npas2, Per2, Cry1, Rorα* and *Reverα*) in the hypothalamus, but not in the retina [84]. Interestingly, the *Rorα* and *RevErbα* gene expressions were phased inversed in parallel with the changes in the activity behavior with the development of *Zugunruhe* in blackheaded buntings.

Experiments on migratory blackheaded bunting have also shown the alteration in daily waveform of clock gene oscillations in both central (pineal, hypothalamus and retina) and peripheral (muscle and liver) tissues [85]. There were changes in both phase and amplitude of 24 h clock gene rhythms as well as in the phase relationship between central and peripheral tissues (Figure 3, [86]). Interestingly, photoperiod-induced seasonal life history dependent changes in circadian clock gene expressions were also found in extra-hypothalamic brain regions [86]. 

#### 3.2.5. Genes Associated with Dopamine Biosynthesis

Vernal migration is goal-directed: males in particular need to arrive at their breeding grounds at a time when they can have a preferential territory and mate choice. This asks for a stronger drive or sexual motivation to depart from their wintering grounds and arrive early at their breeding grounds. Sharma et al. [52] investigated this and found an enhanced hypothalamic mRNA levels of the *Th* gene in migratory redheaded buntings with simulated spring migration phenotype, as compared to those with non-migratory as well as the autumn migratory phenotype (Figure 3). The *Th* gene encodes for tyrosine hydroxylase, which is the rate-limiting enzyme of the dopamine biosynthesis pathway. The reduced *Th* mRNA levels in castrates, compared to intact buntings, further supports this [89]. Interestingly, the expression of *Th* is modulated by the light quality and temperature suggesting a functional interaction of the hypothalamic thermosensitive and photoreceptive cells with dopaminergic neurons [67,90]. Photostimulated redheaded buntings showed higher hypothalamic *Th* mRNA levels when they were exposed to a short light wavelength (460 nm), compared to when they were exposed to a longer light wavelength (620 nm) [90]. Likewise, the exposure to a stimulatory photoperiod at 35 °C led to an increased *Th* mRNA expression, compared to levels at 22 °C, in both the midbrain and hypothalamus of migratory blackheaded buntings (Figure 2, [67]).

#### 3.2.6. Changes in Expression of Genes Linked with Epigenetic Modifications

Distinct seasonal migrations demand genetic plasticity, which is possibly rendered by epigenetic modification processes, such as the DNA-methylation [91,92]. DNA methyltransferases (DNMTs) transfer methyl groups on DNA, whereas the ten-eleven translocation methylcytosine di-oxygenase 2 (TET2) adds a hydroxyl group to the methyl group causing DNA-demethylation [93]. Sharma et al. [52] found differences in the hypothalamic expression of *Dnmt3a* and *Tet2* genes between non-migratory and spring migratory phenotypes of migratory blackheaded buntings (Figure 3). In addition, *Dnmt3a* and *Tet2* mRNA levels were higher in spring and autumn migratory phenotypes, respectively, suggesting a seasonal difference in the methylation process. Another study found differential expressions of the *Dnmt3b* gene between 22 °C and 38 °C temperatures, with higher mRNA levels at 22 °C, in migratory redheaded buntings exposed to stimulatory long days [70]. Furthermore, the *Dio3* expression was negatively and positively correlated with *Dnmt3b* and *Tet2*, respectively, suggesting a possible involvement of epigenetic modification in the induction of a seasonal migration phenotype [70].

#### 3.2.7. Differential Gene Expressions between Spring and Autumn Migrations

As noted above, the spring and autumn migratory travels are distinctly different, and so they probably need differential molecular strategies to achieve their goals. In a series of investigations, Kumar and colleagues investigated this mainly using redheaded buntings as the experimental system (see especially, [52,53]). In particular, there were differences in the diurnal expression pattern of *Per2*, *Cry1,* and *Adcyap1* genes between photostimulated spring and autumn migratory phenotypes, suggesting a differential responsiveness of the light-responsive circadian pathway [52]. The thyroid hormone responsive pathway genes (*Dio2*, *Dio3*) also showed differences in their expression patterns, with higher *Dio2* and *Dio3* mRNA levels in buntings exhibiting spring and autumn migratory phenotypes, respectively [52].

As would be expected with differential metabolic requirements between two migrations (the faster spring migration requiring more energy), buntings exhibiting spring migratory phenotype showed a higher mRNA expression of genes involved in the fat metabolism [53]. For example, buntings exhibiting the spring migratory phenotype had higher mRNA expression of the adipose triglyceride lipase (*Atgl*) gene [53], which is responsible for >90% of triglyceride hydrolysis [94]. The mRNA expression of fatty acid translocase (*Fat*/*Cd36*), fatty acid binding protein (*Fabp3*) and carnitine palmitoyl transferase (*Cpt1*) genes, which are responsible for drawing free fatty acid from the circulation into flight muscle for its β-oxidation and hence energy generation, were also higher in birds in the spring migratory than the autumn migratory state. Increased muscular expression of *Myod1* and *Pvalb* genes further suggested an enhanced myogenesis, hence an increased support to the strength of extensively working flight muscles in spring migration (Figure 3, [53]). 

### 3.3. Global Gene Analyses

Migration is not a single component event; it is rather an outcome of several sequentially occurring component events, viz., the migratory urge, preparedness for the long journey, direction to fly, distance to cover and the mechanism to replenish energy during stopovers. Therefore, instead of a single or a set of genes, the genetic control for migration may involve an entire “migratory gene package”. The constituent genes of such a “gene package” must show differential expression possibly with varying degrees among its candidates during different stages of the migration. The study of such a proposition has become possible through recent technological advancements, such as the next-gen sequencing technique that allows for whole-genome or transcriptome sequencing.

Using the global gene approach, there have been attempts in the last ten years to identify the component genes of the “migratory gene package”. In a first study of the kind, using 454 pyrosequencing, Lundberg et al. [95] compared brain-derived transcriptomes of *P. t. trochilus* and *P. t. acredula* and found 55 highly differentiated SNPs between two subspecies clustering largely to two chromosome regions, possibly influenced by the divergent selection and adaptation to differential migratory strategies [95]. Similarly, Fudickar et al. [96] found 547 differentially expressed genes in peripheral tissues (blood and pectoral muscle) between migratory and sedentary populations of dark-eyed juncos (*Junco hyemalis*). There was an increased expression of genes associated with lipid transport and fatty acid catabolic processes in the muscle and with a ribosomal structure that indicated protein synthesis in the blood of the migratory *J. h. hyemalis*, as compared to those in the sedentary *J. h. carolinensis* [96].

Further studies focused on the differences in gene expressions in between seasonal states of the same species. Using microarrays, Boss et al. [97] reported substantial differences in the number of differentially expressed genes (DEGs, 13.8%), particularly enriching the calcium ion transport, neuronal firing and neuronal synapse formation pathways, in willow warblers’ brains between breeding and autumn migration periods [97]. Similarly, using RNASeq of the ventral hypothalamus region, Johnston et al. [98] reported a higher expression of genes involved in focal adhesion, proliferation and motility in migratory than in the non-migratory state in captive Swainson’s thrushes (*Catharus ustulatus*). On the other hand, Franchini et al. [99] reported only four differentially expressed genes linked with the hyperphagia, moulting and enhanced DNA replication in the blood transcriptome of partial migratory European blackbirds (*Turdus merula*).

More recently, our laboratory has performed a number of transcriptome analyses of the hypothalamus and liver tissues of migratory blackheaded and redheaded buntings. We found an enhanced expression of genes enriching the ATP binding pathway, focal adhesion and intracellular protein transport in the hypothalamus and that of calcium ion transport and small GTPase-mediated signal transduction in the liver of blackheaded buntings exhibiting a photostimulated spring migration state [100]. The RNA-seq of the liver tissue further showed a 24 h cycle in the expression of 4448 genes, and 569 differentially expressed genes between spring migratory and post-migratory states in blackheaded buntings [68]. These differentially expressed hepatic genes were involved in both transcription and post-transcriptional modifications, as required for differential metabolism with the development and cessation of the migratory state [68]. As expected, the genes associated with energy generation (e.g., oxidative phosphorylation and fat metabolism) were highly expressed during the night, while those associated with fat accumulation and refueling were upregulated during the daytime in buntings that were photostimulated into the migratory state. Likewise, genes involved in the recovery pathway such as cell death, gluconeogenesis, etc. show increased expression in the post-migratory state. Most interestingly, there were differences in the gene expression pattern particularly early in the day and night between nonmigratory and migratory states, suggesting differences in the diurnal metabolic support to different seasonal life history states in migratory buntings [101].

## 4. Perspective

The present review is an update on advances in the genetic control of seasonal migration in birds, with a particular focus on latitudinal passerine migrants that outnumber the other groups of avian migrants. It has to be considered that different genetic programs might have evolved to anticipate and tune with several changes in the external environment (abiotic factors) that create a short favorable window and shape the execution of a high-energy seasonal event such as migration. Although the primary abiotic driver of seasonal migration is the change in environmental photoperiod [58], there are also significant influences of temperature [67,70] and food availability [102] on when exactly a species should migrate.

The internal genetic programs involve a host of cellular signaling cascades and operate in sync with each other so that they can effectively integrate and transduce information from the environment to relevant effector organs which are functionally ready when they are needed. For example, the mechanisms underlying timekeeping will create the temporal window (using the circannual timer [103,104]) and alteration in daily behavior (from diurnal to nocturnal activity pattern using the circadian timer [58,82]) while those underlying the energy homeostasis need to be ready to cater the metabolic needs of a migrant species.

We believe that the recent global genetic approach is significantly advancing our knowledge of candidate molecules that possibly underpin the seasonal biological processes, such as migration in birds. However, there is a need for more intense and concerted research on (i) the characterization and differential expression of a host of genes at different stages of annual life history, and (ii) how differentially expressed candidate genes are translated and subjected to post-translational modifications that ultimately govern the biological function. We would like to note further that (i) within and between population genetic variations in migratory traits might not be found in the sequence of differentially expressed genes, but rather in regulatory elements upstream or elsewhere in the genome [105], and (ii) apart from genetics, a complex set of factors including those from the environment (epigenetic control) shape an innate migration template into a seasonally realized migration phenotype [48,103,106,107]. All this makes the avian migration an interesting and exciting field for biological research.

## Figures and Tables

**Figure 1 genes-14-01191-f001:**
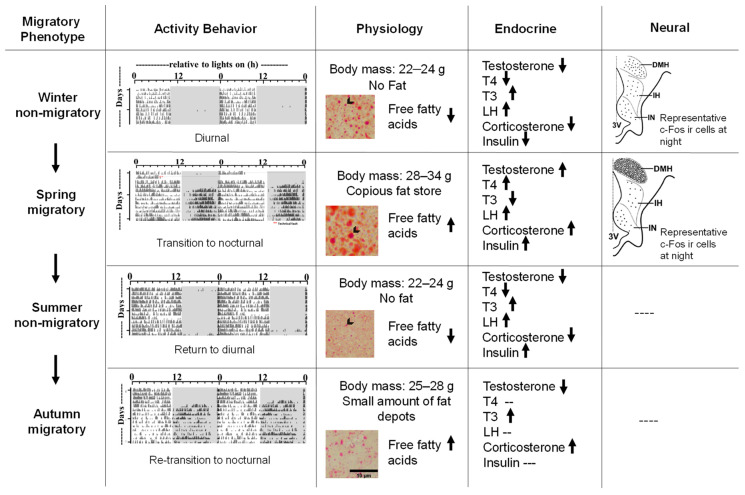
**Schematic illustration of annual life history of latitudinal obligate migratory buntings (*Emberiza* sp.).** Panels (from left to right): Panel on the extreme left shows double plotted representative actograms of captive birds in non-migratory and migratory phenotypes. The white and gray shaded areas represent day and night time, respectively. Note that in non-migratory state, the birds remain active during the day (diurnal), while during migratory state, they become predominantly night active (nocturnal). The second panel lists change in body mass, fat deposition and hepatic neutral lipid accumulation as visualized by Oil Red O staining (magnification: ocular ×10, objective ×40) in different seasonal states. Note the lipid-laden liver cells during the spring migratory state. The next panel summarizes endocrine (hormonal) changes in different seasonal states. The extreme right panel shows a representative diagram of the hypothalamus showing Fos-like immunoreactive cells in the winter non-migratory and spring migratory states. Abbreviation: T4: thyroxine, T3: triiodothyronine, LH: Luteinizing hormone, DMH: dorsomedial hypothalamus, IH: inferior hypothalamic nucleus, IN: Infundibular nucleus, 3V: third ventricle. The figure has been drawn based on findings in several recent publications from our laboratory [52,53,54,55,56].

**Figure 2 genes-14-01191-f002:**
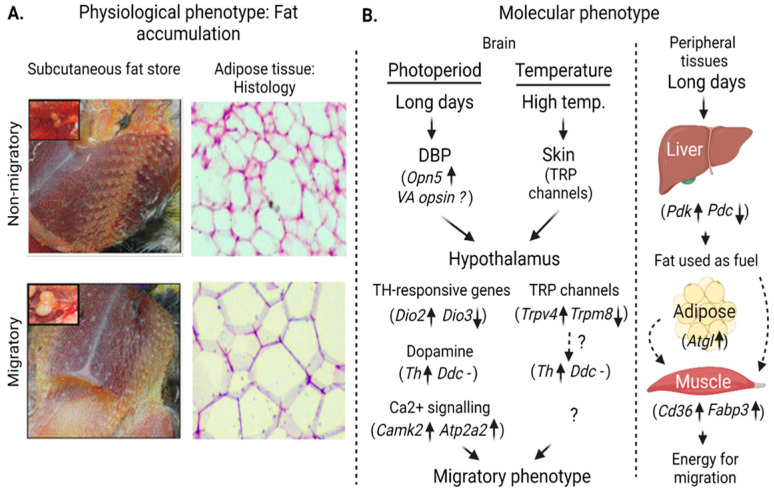
**Schematic diagram depicting changes in physiological and molecular changes with transition from non-migratory to migratory states in captive buntings.** (**A**) Representative images show changes in testes size (inset: **top left**), fat depots around keel region (**left**), and H-E stained histological images of adipose tissue (**right**: magnification: ocular ×10, objective ×10) in photostimulated non-migratory and migratory states. Note larger testes size, more fat depots and larger adipose cells in the migratory state. (**B**) **Left panel**: hypothalamic transduction of increasing photoperiod via DBP (deep brain photoreceptors) and thermal (largely via skin) information and consequently eliciting changes in thyroid hormone-responsive, TRP channel, dopamine biosynthesis and Ca^2+^ signaling genes. **Right Panel**: molecular changes in the liver, adipose tissue and muscle to support enhanced energy requirement during migration. The figure has been drawn based on findings in recent publications from our laboratory [52,53,67,68].

**Figure 3 genes-14-01191-f003:**
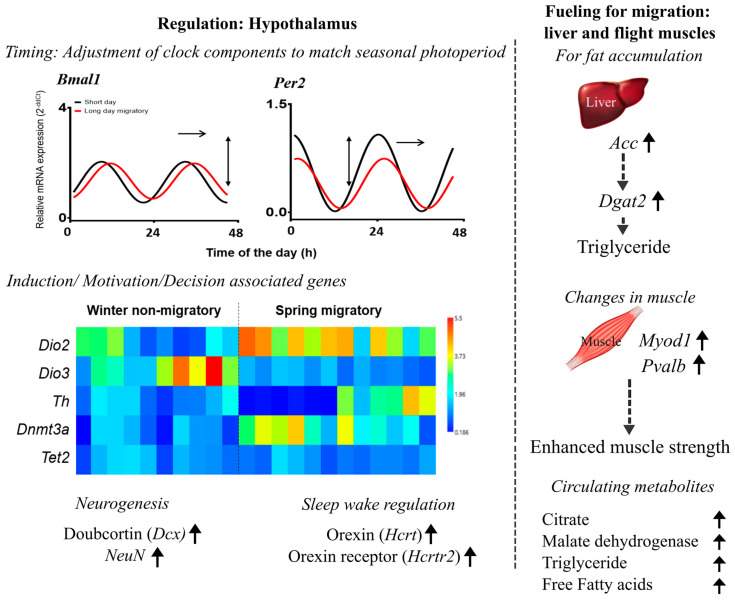
**Summary of gene expression patterns associated with photoperiodic induction of spring migration in captive buntings.** Left panel: mRNA expression of selected gene in the hypothalamus. Graphs on top represent example genes involved in the regulation of migration, for example, changes in phase or amplitude or both in 24 h rhythm in clock genes (*Bmal1* and *Per2*). Similarly, the heatmap compares the expression pattern of five genes between non-migratory and spring migratory state. Here, each column represents one individual for each gene (row). Note that the first five (in non-migratory) or six (in migratory state) columns show the mRNA expression levels during the middle of the day, and the next columns show the mRNA expression levels during the middle of the night. At the bottom are listed genes associated with neurogenesis and sleep wake regulation. An upward arrow suggests their upregulation in spring migratory, compared to the winter non-migratory state. Right panel: Diagrammatic presentation of increased expression of selected genes in the liver associated with fat fueling and flight muscle associated with strength with the development of migratory phenotype. The figure has been drawn based on findings in several recent publications from our laboratory [52,53,85,87,88].

## Data Availability

No new data were created or analyzed in this study. Data sharing is not applicable to this article.

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
