# Peer review of "Genetic Control of Avian Migration: Insights from Studies in Latitudinal Passerine Migrants"

_genes, 2023, doi:10.3390/genes14061191_

Round 1

Reviewer 1 Report

This current review aims to compile state of the art knowledge on molecular mechanisms underlying the genetic basis of songbird migration particularly focusing on gene expression. Certainly, a very interesting topic with recent advances and no recent reviews written about it. There is certainly room and a need for a new review on the topic. Authors have put lot of work in extracting precise data on various molecular pathways and other details from many previous studies. The Authors have over years produced significant amount of research on black headed buntings and it is great to have all these results summed up in this review paper.

However, this draft suffers from a lack of focus and most paragraphs simply list results from many previous studies without any meaningful discussion of what the results mean. This review fails to do what a review should, which is putting all the findings in larger perspective, not simply reciting results from other studies. A review should also provide certain critique of the work that’s been done, as well as attempt to lay some framework for the future. I have listed specific concerns of mine in line-by-line comments.

Studying gene expression is particularly challenging because gene expression patterns may differ significantly at different times of year, different hours of day. Moreover this will differ depending on which type of tissue is sampled. I think these authors have the knowledge to attempt to solve these methodological problems and lay out some recommendations for such studies. If they decide to rewrite the manuscript inclusion of such guidelines would make it very valuable to the research community.        

This review would require major revisions, as of now it is difficult to read and main messages (if any) are lost in overwhelming and not always necessary reports on details of a long list of papers. 

Comments line by line

L 32     “..direction and location of migration …” replace migration with “winter grounds”.

L 41-46 The 2nd and 3rd methods are mentioned but no references given, please provide some.

L 47       Perhaps don’t call it an “event” but maybe behaviour? Event is not heritable, but a behaviour is.

L 69-78 The entire paragraph only credits Anders Pape Moller for reporting results on heritability of spring arrival date. Given the reputation of the said researcher of proven fraud accusations I would recommend being more critical toward these findings in barn swallows. I cannot recall whether there are any other such studies showing genetic control of spring arrival date but I recommend that authors check and see if any other research groups have provided this kind of information.

L 98-103             This paragraph is obsolete and unnecessarily explains very outdated methods such as AFLP and microsatellites. This has nothing to do with the focus of the review which is gene expression. I’d suggest deleting these lines.

L 103     Reference 31 does not say anything about gene expression.

L 105     Is it necessary to explain what a genetic polymorphism is in a paper for a specialist journal?

              Perhaps call it genetic variation?

L 107     What do you mean by “(e.g. a reference experiment)”. Did you forget to add a reference here?

L 110-116           The comparison with Fidler study is irrelevant. It’s a genotyping study that finds some association between variation in some loci with exploratory behaviour in great tits. I see no reason why this is relevant to migration.

              This paragraph discusses the importance of the Clock gene, however there is important critique missing. Please see the reference below. This study does a proper examination of all reports of the poly q allele length and convincingly shows that variation at this locus has no control over migration timing.

Parody-Merino, Á.M., Battley, P.F., Conklin, J.R. et al. No evidence for an association between Clock gene allelic variation and migration timing in a long-distance migratory shorebird (Limosa lapponica baueri). Oecologia 191, 843–859 (2019). https://doi.org/10.1007/s00442-019-04524-8

A meaningful paper you should discuss here is Bossu et al https://doi.org/10.1098/rspb.2021.2507 which reports on core clock genes being conserved and regulatory loci being polymorphic. This variation does seem to affect migration timing in American kestrels.

The literature citations for this section are very incomplete and lack several recent and important papers:

              Delmore et al https://doi.org/10.7554/eLife.54462

              Delmore et al http://dx.doi.org/10.1016/j.cub.2016.06.015

              Sokolovskis et al https://doi.org/10.1038/s41467-023-35788-7

              Gu et al https://doi.org/10.1038/s41586-021-03265-0

              Caballero-Lopez et al DOI: 10.1111/mec.16292

              Since the focus of the review is gene expression, I think this section can be trimmed away, unless you want to substantially expand it and mention proper citations.

L 152     Since you say “Several such studies” please add citations.

Figure 1    The activity data patterns look different than in the reference 70? Are you sure they are from the same study.

 Figure legend (L 171-172) you should refer exactly which data was from which study. As it is now you say that the data is “especially” from these studies. This is not a precise language and doesn’t allow reader who might be interested to find relevant published papers and or deposited data sets.

L 193    “Long day breeding birds” is a confusing wording. Please write it out more understandably.

Figure 2   What do the adipose tissue images in the panel A mean? There seem to be some differences but no explanation of what is depicted. Is it that the bottom migratory state slide shows enlarged cells? If they are enlarged relative to the slide above there should be some scale of size.

L 218     Same as figure 1, “see especially”. You need to be specific with referring to source of data for the figures.

L 239     I don’t think birds with more fat reserves migrate faster. Did you perhaps mean that birds that have accumulated more fat are more likely to depart sooner than lean birds that haven’t accumulated enough fat yet?

Figure 3 L 316. Same as previous figures, be precise when referring to which panel contains information from which paper.

L 317-333           I think saying genes associated with motivation is too vague. I certainly do not doubt that there must be some rewarding stimuli for birds to start and carry on migration. However the discussed expression differences in the Th gene as you say in lines 327-328, are modulated by light and temperature. The link with Th expression levels and motivation seems very arbitrary.

L 334 -348          This section is supposed to deal with epigenetics and methylation patterns. However, the only studies discussed here are on different expression patterns of some selected genes. I don’t see the link between the mentioned genes and methylation. In principle any gene can be affected by epigenetic modifications, but until proven its just a guess.

L 376     Birds do not replenish energy while in flight but between flights on stopovers.

Paragraph 3.3 as several others in the manuscript simply reports results of different studies. Such report could fit an informative supplementary material table. Discussion of the results and the meaning of the lack similar results or presence of similar results between species is what should be discussed here.

           L 433     This is not a brief update but a rather extensive one.

L 436     I do not think that anybody in this research field thinks that molecular mechanisms underlying migration are simplistic.

L 442     There is no reason to think that food and change in environmental conditions dictate the birds where to migrate. 

The text should be carefully proof checked by a native speaker. Spelling seems to be fine but usage of awkward words in many instances makes the text difficult to understand. 

Author Response

Response to Reviewer 1 Comments

Comment #1: This current review aims to compile state of the art knowledge on molecular mechanisms underlying the genetic basis of songbird migration particularly focusing on gene expression. Certainly, a very interesting topic with recent advances and no recent reviews written about it. There is certainly room and a need for a new review on the topic. Authors have put lot of work in extracting precise data on various molecular pathways and other details from many previous studies. The Authors have over years produced significant amount of research on black headed buntings and it is great to have all these results summed up in this review paper.

However, this draft suffers from a lack of focus and most paragraphs simply list results from many previous studies without any meaningful discussion of what the results mean. This review fails to do what a review should, which is putting all the findings in larger perspective, not simply reciting results from other studies. A review should also provide certain critique of the work that’s been done, as well as attempt to lay some framework for the future. I have listed specific concerns of mine in line-by-line comments. 

Response #1: Thank you. We have addressed concerns raised in the revised version. 

Comment #2: Studying gene expression is particularly challenging because gene expression patterns may differ significantly at different times of year, different hours of day. Moreover, this will differ depending on which type of tissue is sampled. I think these authors have the knowledge to attempt to solve these methodological problems and lay out some recommendations for such studies. If they decide to rewrite the manuscript inclusion of such guidelines would make it very valuable to the research community.

Response #2: Thank you. Included (Lines 185-190)

Comment #3: This review would require major revisions, as of now it is difficult to read and main messages (if any) are lost in overwhelming and not always necessary reports on details of a long list of papers. 

Response #3: Thank you. Revised the text.

Comment #4: L 32 “..direction and location of migration …” replace migration with “winter grounds”.

Response #4: Corrected (Line 49).

Comment #5: L 41-46 The 2nd and 3rd methods are mentioned but no references given, please provide some.

Response #5: Included (Lines 59 and 61).

Comment #6: L 47       Perhaps don’t call it an “event” but maybe behaviour? Event is not heritable, but a behaviour is.

Response #6: Corrected (Line 62).

Comment #7: L 69-78 The entire paragraph only credits Anders Pape Moller for reporting results on heritability of spring arrival date. Given the reputation of the said researcher of proven fraud accusations I would recommend being more critical toward these findings in barn swallows. I cannot recall whether there are any other such studies showing genetic control of spring arrival date but I recommend that authors check and see if any other research groups have provided this kind of information.

Response #7: Thank you for pointing this out. We have included other references with similar findings and removed A. P. Moller’s study (Line 110-114).

Comment #8: L 98-103 This paragraph is obsolete and unnecessarily explains very outdated methods such as AFLP and microsatellites. This has nothing to do with the focus of the review which is gene expression. I’d suggest deleting these lines.

Response #8: Deleted.

Comment #9: L 103 Reference 31 does not say anything about gene expression.

Response #9: Thank you. Corrected.

Comment #10: L 105     Is it necessary to explain what a genetic polymorphism is in a paper for a specialist journal?Perhaps call it genetic variation?

Response #10: Thank you. Reworded.

Comment #11: L 107     What do you mean by “(e.g. a reference experiment)”. Did you forget to add a reference here?

Response #11: Thank you. Reworded.

Comment #12: L 110-116 The comparison with Fidler study is irrelevant. It’s a genotyping study that finds some association between variation in some loci with exploratory behaviour in great tits. I see no reason why this is relevant to migration.

Response #12: Thank you. Reworded the segment.

Comment #13: This paragraph discusses the importance of the Clock gene, however there is important critique missing. Please see the reference below. This study does a proper examination of all reports of the poly q allele length and convincingly shows that variation at this locus has no control over migration timing.

Parody-Merino, Á.M., Battley, P.F., Conklin, J.R. et al. No evidence for an association between Clock gene allelic variation and migration timing in a long-distance migratory shorebird (Limosa lapponica baueri). Oecologia 191, 843–859 (2019). https://doi.org/10.1007/s00442-019-04524-8

Response #13: Thank you. The text is revised (Lines 144-146).

Comment #14: A meaningful paper you should discuss here is Bossu et al https://doi.org/10.1098/rspb.2021.2507 which reports on core clock genes being conserved and regulatory loci being polymorphic. This variation does seem to affect migration timing in American kestrels. The literature citations for this section are very incomplete and lack several recent and important papers:

Delmore et al https://doi.org/10.7554/eLife.54462

Delmore et al http://dx.doi.org/10.1016/j.cub.2016.06.015

Sokolovskis et al https://doi.org/10.1038/s41467-023-35788-7

Gu et al https://doi.org/10.1038/s41586-021-03265-0

Caballero-Lopez et al DOI: 10.1111/mec.16292

Since the focus of the review is gene expression, I think this section can be trimmed away, unless you want to substantially expand it and mention proper citations.

Response #14: Thank you. We would still like to retain this section and so we have included the suggested literature and discussed the relevant findings in detail (Lines 150-171). 

Comment #15: L 152     Since you say “Several such studies” please add citations.

Response #15: Thank you. Included (Line 193)

Comment #16: Figure 1 The activity data patterns look different than in the reference 70? Are you sure they are from the same study.

Response #16: Thank you. Yes, the activity pattern are representative actograms from different individuals of the same study.

 Comment #17: Figure legend (L 171-172) you should refer exactly which data was from which study. As it is now you say that the data is “especially” from these studies. This is not a precise language and doesn’t allow reader who might be interested to find relevant published papers and or deposited data sets.

Response #17: Thank you. Included (Fig. legend 1)

Comment #18: L 193 “Long day breeding birds” is a confusing wording. Please write it out more understandably.

Response #18: Thank you. Modified as “long day breeders” (Line 214)

Comment #19: Figure 2   What do the adipose tissue images in the panel A mean? There seem to be some differences but no explanation of what is depicted. Is it that the bottom migratory state slide shows enlarged cells? If they are enlarged relative to the slide above there should be some scale of size.

Response #19: Thank you. Details included in the revised figure legend 2.

Comment #20: L 218 Same as figure 1, “see especially”. You need to be specific with referring to source of data for the figures.

Response #20: Thank you. Included (Fig. legend 2)

Comment #21: L 239 I don’t think birds with more fat reserves migrate faster. Did you perhaps mean that birds that have accumulated more fat are more likely to depart sooner than lean birds that haven’t accumulated enough fat yet?

Response #21: Thank you. Reworded (Lines 240-244).

Comment #22: Figure 3 L 316. Same as previous figures, be precise when referring to which panel contains information from which paper.

Response #22: Thank you. Included (Fig. legend 3)

Comment #23: L 317-333 I think saying genes associated with motivation is too vague. I certainly do not doubt that there must be some rewarding stimuli for birds to start and carry on migration. However, the discussed expression differences in the Th gene as you say in lines 327-328, are modulated by light and temperature. The link with Th expression levels and motivation seems very arbitrary.

Response #23: Thank you. We have re-worded the statement to make it clearer (Line 303). How the modulation of Th by light and temperature may affect the motivation is also discussed (Line 313).

Comment #24: L 334 -348 This section is supposed to deal with epigenetics and methylation patterns. However, the only studies discussed here are on different expression patterns of some selected genes. I don’t see the link between the mentioned genes and methylation. In principle any gene can be affected by epigenetic modifications, but until proven its just a guess.

Response #24: Thank you. Reworded based on the current available literature (Line 320)

Comment #25: L 376 Birds do not replenish energy while in flight but between flights on stopovers.

Response #25: Thank you. Corrected (Line 360).

Comment #26: Paragraph 3.3 as several others in the manuscript simply reports results of different studies. Such report could fit an informative supplementary material table. Discussion of the results and the meaning of the lack similar results or presence of similar results between species is what should be discussed here.

Response #26: Thank you. Revised but it is felt that it is better to be retained in the main text (Lines 369-378).

Comment #27: L 433 This is not a brief update but a rather extensive one.

Response #27: Thank you. Corrected (Line 410).

Comment #28: L 436 I do not think that anybody in this research field thinks that molecular mechanisms underlying migration are simplistic.

Response #28: Thank you. Reworded.

Comment #29: L 442 There is no reason to think that food and change in environmental conditions dictate the birds where to migrate. 

Response #29: Thank you. Reworded.

Reviewer 2 Report

The study is an inetresting review of recent findings in terms of genetic mechanisms of migration in birds, based on the series of studies of two species of long-distance migrant buntings between Palaearctic and India. I found the paper a helpful, clear and wide overview of findings considering a veriety of physiological changes in different organs, tissues, hormonal excretion, and the genetic basis of their expressions. I found only a few minor flaws in the paper, in fragments considering more general information on birds’ migrations.

1)      Line 69, paragraph 2.2: I  suggest that in this paragraph, and also in further parts of the manuscript, the authors refers to more recent studies in the field of genetics of bird migration, for example:

1.Merlin, C., & Liedvogel, M. (2019). The genetics and epigenetics of animal migration and orientation: birds, butterflies and beyond. Journal of Experimental Biology, 222(Suppl_1), jeb191890.

2.Caballero‐López, V., Lundberg, M., Sokolovskis, K., & Bensch, S. (2022). Transposable elements mark a repeat‐rich region associated with migratory phenotypes of willow warblers (Phylloscopus trochilus). Molecular Ecology, 31(4), 1128-1141.

3. Sokolovskis, K., Lundberg, M., Åkesson, S., Willemoes, M., Zhao, T., Caballero-Lopez, V., & Bensch, S. (2023). Migration direction in a songbird explained by two loci. Nature Communications, 14(1), 165.

4. Zhao, T., Ilieva, M., Larson, K., Lundberg, M., Neto, J. M., Sokolovskis, K., ... & Bensch, S. (2020). Autumn migration direction of juvenile willow warblers (Phylloscopus t. trochilus and P. t. acredula) and their hybrids assessed by qPCR SNP genotyping. Movement ecology, 8, 1-10.

2) Line 236 „Birds accumulate fat (fuel), which is greatest in small birds….”

This statement is not quite true, and the quoted reference is rather old (1960). Newer studies show that the records in terms of fat deposits belong rather to medium-sized waders, which are record fliers. Please, rephrase this statement and provide newer reference.

3) I suggest that the figure captions are shorter, and thus easier to comprehend, I think they can be written more succintly.

I include these comments, and a number of smaller editorial comments, in the attached manuscript.

I suggest a few minor corrections.

Author Response

Response to Reviewer 2

Comment #1: The study is an inetresting review of recent findings in terms of genetic mechanisms of migration in birds, based on the series of studies of two species of long-distance migrant buntings between Palaearctic and India. I found the paper a helpful, clear and wide overview of findings considering a veriety of physiological changes in different organs, tissues, hormonal excretion, and the genetic basis of their expressions. I found only a few minor flaws in the paper, in fragments considering more general information on birds’ migrations. 

Response #1: Thank you. We have included all the suggestions in the revised version.

Comment #2: Line 69, paragraph 2.2: I  suggest that in this paragraph, and also in further parts of the manuscript, the authors refers to more recent studies in the field of genetics of bird migration, for example: 

1.Merlin, C., & Liedvogel, M. (2019). The genetics and epigenetics of animal migration and orientation: birds, butterflies and beyond. Journal of Experimental Biology, 222(Suppl_1), jeb191890.

2.Caballero‐López, V., Lundberg, M., Sokolovskis, K., & Bensch, S. (2022). Transposable elements mark a repeat‐rich region associated with migratory phenotypes of willow warblers (Phylloscopus trochilus). Molecular Ecology, 31(4), 1128-1141.

  1. Sokolovskis, K., Lundberg, M., Åkesson, S., Willemoes, M., Zhao, T., Caballero-Lopez, V., & Bensch, S. (2023). Migration direction in a songbird explained by two loci. Nature Communications, 14(1), 165.
  2. Zhao, T., Ilieva, M., Larson, K., Lundberg, M., Neto, J. M., Sokolovskis, K., ... & Bensch, S. (2020). Autumn migration direction of juvenile willow warblers (Phylloscopus t. trochilus and P. t. acredula) and their hybrids assessed by qPCR SNP genotyping. Movement ecology, 8, 1-10.

Response #2: Thank you. Included (Lines 90-105).

Comment #3: Line 236 Birds accumulate fat (fuel), which is greatest in small birds….” This statement is not quite true, and the quoted reference is rather old (1960). Newer studies show that the records in terms of fat deposits belong rather to medium-sized waders, which are record fliers. Please, rephrase this statement and provide newer reference. 

Response #3: Thank you. Reworded (Line 241).

Comment #4: I suggest that the figure captions are shorter, and thus easier to comprehend, I think they can be written more succintly. 

Response #4: Thank you. Reworded